# Characterization of cerebrospinal fluid (CSF) microbiota from patients with CSF shunt infection and reinfection using high throughput sequencing of 16S ribosomal RNAgenes

Kathryn B. Whitlock[1], Christopher E. Pope[2], Paul Hodor[3], Lucas R. Hoffman[2,3], David L. Limbrick, Jr.[4,5], Patrick J. McDonald[6,7], Jason S. Hauptman[3,8], Jeffrey G. Ojemann[3,8], Tamara D. Simon[9,10]*, on behalf of the Cerebrospinal FLuId MicroBiota in Shunts Study (CLIMB) Group¶

1 New Harmony Statistical Consulting LLC, Shoreline, Washington, United States of America, 2 Department of Pediatrics, University of Washington, Seattle, Washington, United States of America, 3 Seattle Children's Research Institute, Seattle, Washington, United States of America, 4 Department of Neurosurgery, Washington University in St. Louis, St. Louis, Missouri, United States of America, 5 St. Louis Children's Hospital, St. Louis, Missouri, United States of America, 6 Division of Neurosurgery, University of British Columbia, Vancouver, British Columbia, Canada, 7 British Columbia Children's Hospital, Vancouver, British Columbia, Canada, 8 Department of Neurosurgery, University of Washington, Seattle, Washington, United States of America, 9 Department of Pediatrics, Keck School of Medicine at the University of Southern California, Los Angeles, California, United States of America, 10 Children's Hospital Los Angeles and The Saban Research Institute, Los Angeles, California, United States of America

¶ Membership of the CLIMB Group is provided in the Acknowledgments.
* tsimon@chla.usc.edu

## Abstract

### Background

Nearly 20% of patients with cerebrospinal fluid (CSF) shunt infection develop reinfection. It is unclear whether reinfections are caused by an organism previously present or are independent infection events.

### Objective

We used bacterial culture and high throughput sequencing (HTS) of 16S ribosomal RNA (rRNA) genes to identify bacteria present in serial CSF samples obtained from children who failed CSF shunt infection treatment. We hypothesized that organisms that persist in CSF despite treatment would be detected upon reinfection.

### Design/methods

Serial CSF samples were obtained from 6 patients, 5 with 2 infections and 1 with 3 infections; the study was limited to those for which CSF samples were available from the end of infection and beginning of reinfection. Amplicons of the 16S rRNA gene V4 region were

**Data Availability Statement:** 16S rRNA sequencing data is available from NCBI Sequence Read Archive PRJNA608523.

**Funding:** This study was supported by the National Institutes of Health via R01 NS095979 awarded to TDS, CEP, PH, LRH, DLL, PM, and KW via New Harmony Statistical Consulting LLC. DLL receives research funding and equipment for unrelated projects through Microbot Medical, Inc. and Medtronic, Inc. LRH receives additional support from the National Institutes of Health via K24 HL141669. The funder provided support in the form of salaries for authors TDS, CEP, PH, LRH, DLL, PM, and KW via New Harmony Statistical Consulting LLC, but did not have any additional role in the study design, data collection and analysis, decision to publish, or preparation of the manuscript. The specific roles of these authors are articulated in the 'author contributions' section.

**Competing interests:** This study was supported by the National Institutes of Health via R01 NS095979 awarded to TDS, CEP, PH, LRH, DLL, PM, and KW via New Harmony Statistical Consulting LLC. This does not alter our adherence to PLOS ONE policies on sharing data and materials.

sequenced. Taxonomic assignments of V4 sequences were compared with bacterial species identified in culture.

## Results

Seven infection dyads averaging 13.5 samples per infection were analyzed. A median of 8 taxa [interquartile range (IQR) 5–10] were observed in the first samples from reinfection using HTS. Conventional culture correlated with high abundance of an organism by HTS in all but 1 infection. In 6 of 7 infection dyads, organisms identified by culture at reinfection were detected by HTS of culture-negative samples at the end of the previous infection. The median Chao-Jaccard abundance-based similarity index for matched infection pairs at end of infection and beginning of reinfection was 0.57 (IQR 0.07–0.87) compared to that for unmatched pairs of 0.40 (IQR 0.10–0.60) [p = 0.46].

## Conclusion(s)

HTS results were generally consistent with culture-based methods in CSF shunt infection and reinfection, and may detect organisms missed by culture at the end of infection treatment but detected by culture at reinfection. However, the CSF microbiota did not correlate more closely within patients at the end of infection and beginning of reinfection than between any two unrelated infections. We cannot reject the hypothesis that sequential infections were independent.

## Introduction

Cerebrospinal fluid (CSF) shunt placement allows children with hydrocephalus, a common cause of neurological disability in children [1], to survive and avoid ongoing brain injury. However, CSF shunts can cause new chronic surgical problems [2]. After initial placement, CSF shunts often require surgical revision for mechanical failure. Revisions are the major risk factor for subsequent CSF shunt infections [3, 4].

Diagnosis of CSF shunt infection currently relies on the recovery of a pathogen from conventional microbiologic cultures of CSF, and duration of antibiotic treatment is often determined based on the organism and the number of days that cultures remain positive. The most common pathogens recovered from conventional microbiologic cultures include primarily *Staphylococcus epidermidis* and *Staphylococcus aureus*, followed by gram-negative organisms. [5] CSF shunt infection treatment generally requires surgical removal of the CSF shunt, one to two weeks of intravenous antibiotics tailored to the organism recovered from conventional culture, and a second surgery to place a new CSF shunt [5, 6]. CSF shunt replacement generally does not occur until negative CSF cultures are obtained and prescribed treatment is complete. Despite this aggressive treatment, prognosis for clearance is poor for children with first CSF shunt infection, with re-infection rates ranging from 20 to 25% [5, 7, 8]. Re-infection rates are higher still for children with their second CSF shunt infection [8]. An improved understanding of the mechanisms of infection is critical to effectively treat the over 2,000 CSF shunt infections diagnosed each year in the United States [9].

Research by our group and others has identified surprisingly few patient, medical, or surgical risk factors associated with either a first or subsequent re-infection [3, 4, 10]. Indirect evidence suggests that CSF shunt infections could commonly involve microbes not reliably detected by

conventional bacterial cultures but that can be identified by molecular microbiological tools, such as high throughput sequencing and quantitative PCR (qPCR) [11]. Previously we used these methods to characterize the DNA of culturable and non-culturable microbes detected in CSF shunt infection [12] and to quantify total CSF bacterial loads [13]. These earlier studies suggested the common presence of low levels of diverse bacterial and fungal DNA in the CSF of children with shunt infection even in the absence of detection by culture. Since then, advances in these molecular approaches, and the laboratory and computational pipelines involved, have markedly increased both sensitivity and specificity, including more rigorous identification and exclusion of contaminants and improved limits of detection [14].

Among the 20 to 25% of patients with CSF shunt infection who develop reinfection, it is unclear whether reinfections are generally caused by organism(s) previously present or are independent infection events. In the majority (70%) of re-infections, the organism(s) recovered by culture differs from those recovered at first infection [8]. We hypothesized that organisms that persist in CSF, undetected by culture, despite treatment would be present in reinfection. To test this hypothesis, we performed HTS of serial CSF samples obtained from children who failed treatment for CSF shunt infection and compared these results to bacterial cultures.

## Materials and methods

### Study subjects

Children ≤18 years old undergoing treatment for conventional culture-confirmed CSF shunt infection at either Seattle Children's Hospital (SCH) in Seattle, Washington or Primary Children's Hospital (PCH) in Salt Lake City, Utah were eligible for enrollment in this study. Enrollment occurred from 2010 to present at SCH and from 2008 to 2015 at PCH. A CSF shunt infection was defined as identification of organisms on microbiological culture of CSF fluid obtained from a partial or complete CSF shunt system. CSF shunt system(s) included ventriculoperitoneal, ventriculoatrial, ventriculopleural, arachnoid cyst shunts, subdural shunts, and lumboperitoneal shunts; temporary devices only such as external ventricular drain(s), Ommaya reservoir(s), ventricular access devices (reservoirs) and subgaleal shunts were excluded. For this study, we examined serial CSF obtained from the subset of children who failed treatment for CSF shunt infection (i.e. had CSF shunt reinfection). We analyzed serial CSF samples from 6 patients, 5 with 2 infections and 1 with 3 infections; the study was limited to those for which CSF samples were available from the end of infection and beginning of reinfection.

### Ethics statement

The study received Institutional Review Board approval from the Seattle Children's Research Institute and the University of Utah, as well as approval from the Primary Children's Hospital Privacy Board. For all study subjects, except those from Primary Children's Hospital prior to March 18, 2010, written consent was obtained from parents or guardians, and assent when age- and developmentally-appropriate from study subjects, for additional CSF to be collected on each occasion that regular CSF samples were obtained during treatment for CSF shunt infection. Prior to March 18, 2010 at Primary Children's Hospital, we used CSF remaining for after routine processing and testing in the Primary Children's Hospital Microbiology Laboratory.

### CSF specimen collection

Sterile conditions were standard practice throughout recovery and storage of CSF. The first CSF sample for diagnosis of infection was usually obtained from needle aspiration of the shunt reservoir under sterile conditions outside the operating room in a bedside "shunt tap". The

initial CSF sample analyzed in this study either was left over from this first diagnostic sample or was obtained in the operating room under sterile conditions from the system being removed during the first surgery to treat infection. Subsequent CSF samples, including those at the end of the infection, were generally obtained under sterile bedside conditions through a sampling port within sterile extension tubing attached to the external ventricular drain.

After CSF was obtained for the study, samples were stored at 4˚C for up to 5 days. CSF was then aliquoted into vials of ~100 μl for the study and stored at -70˚C; PCH samples were shipped overnight to Seattle on dry ice for analysis. After December 9, 2011, 200 μl of CSF was stored in 450 μl MO-BIO CB1 solution (BiOstic®) at the time of sample collection.

## Conventional culture identification of bacteria

All CSF samples were tested using routine aerobic culture techniques in hospital-certified laboratories at both SCH and PCH. Conventional cultures are the traditional diagnostic approach used to detect typical pathogens in infectious diseases and were performed in a clinical microbiology laboratory following Clinical and Laboratory Standards Institute guidelines; however, conventional culture approaches do not detect all bacteria present in human disease [15, 16].

## Clinical data

For each child undergoing CSF shunt infection treatment we collected data about (1) details of surgical treatment (including dates and times, approaches taken, use of neuroendoscope, use of ultrasound, case duration, shunt type, antibiotic impregnated catheter tubing use); (2) antibiotic treatment (including each antibiotic given, amount, route, and numbers of doses); (3) all culture information (including dates, organism recovered if applicable, timing of growth, and antimicrobial sensitivities); and (4) CSF lab results, including CSF culture results and additional CSF tests usually performed with CSF cultures, including CSF white blood cell count, red blood cell count, glucose, protein, and Gram stain.

## DNA extraction

DNA from CSF samples and extraction controls was purified using the AGOWA mag Mini DNA isolation kit (AGOWA, LGC Genomics, Berlin, Germany) in accordance with the manufacturer's recommendations and with modifications. In brief, for each sample a 100 μl volume of CSF was aliquoted into a sterile low binding microfuge tube (Axygen, Catalogue Number (CN): 31104081), to which 20 μl of 20mg/mL Proteinase K (Invitrogen, CN: 25530–049) was added. The mixture was vortexed for 20 seconds and incubated at 55˚C for 10 minutes. After incubation 250 μl of Lysis buffer was added to the tube and vortexed gently for 15 seconds. The mixture was transferred to a clean 2 ml tube (Sarstedt, CN: 72.693.005) containing 0.3 g of 0.1mm zirconia/silica beads (Biospec Products Bartlesville, OK, USA, (Biospec) CN: 11079101z).

Using a Mini-Beadbeater-16 (Biospec, CN: 607) the sample was mechanically disrupted by bead-beating for 3 minutes, followed by a 1-minute incubation on ice and a further 3 minutes of bead-beating. The sample was centrifuged at 4,000 rpm for 10 minutes. The resulting supernatant was transferred to a new low binding microfuge tube. To this, 325 μl of Binding buffer and 10 μl of mag particle suspension (mag-particles) were added, vortexed for 15 seconds, and incubated at room temperature for 30 minutes with continuous mixing on a VWR Tube Rotator (VWR, CN: 10136–084). After the incubation step the tube was placed in a Microfuge Tube Magnetic Separation Rack (mag-rack) (Permagen, CN: MSR24) at room-temperature for 1 minute to pellet the mag-particles. The supernatant was discarded. The mag-particles were resuspended in 200 μl Wash buffer 1 and incubated at room-temperature for 5 minutes

with continuous mixing. The tube was placed in the mag-rack and incubated for 1 minute at room-temperature to allow the mag-particles to form a pellet. The supernatant was discarded. The mag-particles were resuspended in 200 μl Wash buffer 2. The tube was again incubated at room-temperature for 5 minutes with continuous mixing. The tube was placed in the mag-rack and incubated for 1 minute at room temperature to pellet the mag-particles. The supernatant was discarded and the mag-particle pellet was dried by incubating the open tube at 55˚C for 5 minutes. DNA was eluted by adding 63 μl of Elution buffer to the dried mag-particle pellet and mixing thoroughly. The mixture was incubated at 55˚C for 10 minutes with periodic gentle vortexing. The liquid was cooled by placing the tube on ice for 5 minutes. The mag-particles were pelleted by placing the tubes on the mag-rack and incubated for 3 minutes. A 54 μl volume of the eluate was transferred to a sterile low binding microfuge tube and stored at -80˚C until required.

Controls included a pure culture of *Staphylococcus aureus*, duplicate mock community samples (Table 1), and four kit no-sample controls (one for each kit used), each of which underwent DNA extraction and subsequent testing concurrently with study samples as described below.

## Bacterial quantification

A quantitative PCR (qPCR) assay targeting the 16S rRNA genes was used to measure the total bacterial load in each CSF sample. An in-house *Streptococcus mitis* strain was used as the source of DNA for the assay's standard curve. *S. mitis* was grown on Trypticase Soy Agar with defibrinated sheep blood and incubated for 16 hours at 37˚C. Afterwards DNA was purified using the Qiagen DNeasy Blood & Tissue Kit (Qiagen, CN: 69506) following the manufacturer's protocol for Gram-positive bacteria. The standard curve was run in duplicate with each qPCR assay at a 1:10 dilution series of 2000 pg/μl to 0.2 pg/μl of *S. mitis* genomic DNA. This was equivalent to $8.24 \times 10^5$ to $8.24 \times 10^1$ genome equivalents per microliter of DNA (GE/μl).

Quantification of the total bacterial load in CSF samples was carried out in duplicate, with positive and negative controls and a standard curve to calculate the genome equivalents per milliliter (GE/ml) of bacteria in the original CSF sample. The qPCR was carried out using the PowerUP™ SYBR™ Green Master Mix (Applied Biosystems, A25742) reaction mixture. The 20 μL qPCR reaction included 10 μL of PowerUP™ SYBR™ Green Master Mix, 0.5 uM of each primer (qPCR-16F: 5′–TCCTACGGGAGGCAGCAGT–3′ and qPCR-16R: 5′–GGACTACCAGG GTATCTAATCCTGTT–3′) [17], and 1 μL of DNA template, the remaining volume was reached using sterile DNA free water. The reaction was performed on a CFX96 Touch real-time PCR detection system (Biorad) using the following reaction conditions; initial incubation at 95˚C for 4 minutes, followed by 39 cycles of 95˚C for 10 seconds and 60˚C for 1 minute.

**Table 1. Composition of bacterial mock community.**

| Genera | Mock input (Relative Abundance in %) |
|---|---|
| *Escherichia/Shigella* | 1.99 |
| *Haemophilus* | 20.41 |
| *Moraxella* | 19.84 |
| *Prevotella* | 4.67 |
| *Propionibacterium* | 1.63 |
| *Pseudomonas* | 14.62 |
| *Staphylococcus* | 14.99 |
| *Streptococcus* | 21.84 |

The DNAs extracted from the series dilutions were used to set up a standard curve, the Ct values of the quantitative real-time PCR assay (qPCR) assay are compared with the standard curve. The qPCR assay showed that a concentration as low as $10^2$ GE/μl can be detected on the DNA samples.

## Bacterial 16S rRNA gene amplification, sequencing and analyses (high throughput sequencing)

CSF microbiota amplicon library construction was carried out using a one-step PCR amplification targeting the V4 region of the bacterial 16S rRNA gene. The amplicon library primer set [18] each contained the specific Illumina adapters, an 8-bp index sequence to allow for multiplex sequencing of the samples, and the 16S rRNA gene specific primer [19]. PCR was carried out in a total reaction volume of 25 μL containing (final concentrations); 1X SuperFi™ Buffer, dNTP mix (0.2 uM of each), 0.4uM of each primer, 0.5U Platinum™ SuperFi™ DNA Polymerase, 2 μL of CSF DNA template and the final volume made up with molecular grade water. Each sample was amplified on a SimpliAmp thermo cycler (Applied Biosystems CN: A24811) in triplicate with an initial incubation at 94˚C for 3 minutes followed by 30 cycles of 94˚C for 45 seconds, 55˚C for 1 minute, and then 72˚C for 90 seconds. Ten negative (i.e. no-template) control analyses were performed, in triplicate, in conjunction with CSF sample batches. After PCR the triplicate amplicons for each sample were pooled and stored at -80˚C until required.

Prior to sequencing individual amplicon libraries were pooled in equal volume. The pooled 16S rRNA gene libraries were electrophoresed and visualized on a 0.7% agarose gel and purified using the PureLink™ Quick Gel Extraction Kit (Invitrogen, CN: K210012) according to the manufacturer's instructions. Sequencing of the pooled libraries was carried out for 600 cycles on an Illumina MiSeq desktop sequencer using the Miseq Reagent Kit v3.

## Sequencing analyses

Sequencing data were analyzed using the denoising program DADA2 [20] (version1.6.0) as described [21], and aligned to the SILVA 16S reference database (v. 132) [22] to produce a 16S amplicon taxa table for downstream computational analysis. Two strategies were used to address contaminant sequences. Using decontam R package installed from GitHub (https://github.com/benjjneb/decontam) [23], contaminant sequences were identified as those with decreasing frequency at higher concentrations (threshold p < 0.10) or that were more prevalent in kit control samples than in CSF samples (threshold p < 0.50) and removed. Alternately, sequences detected in extraction kit controls were identified as contaminants and were removed from the CSF samples.

## Data analyses

Taxonomic assignments of V4 sequences were compared with bacterial species identified in culture. Microbiota was characterized within patients and across infection episodes; and compared to evaluate preliminary predictive signals to further design a study of CSF shunt infection. Sequence incidence was calculated for each sample and summarized across samples. While the analysis was conducted at sequence level, Figs 1, 3 and 4 present results at the genus level for ease of interpretability. S1 and S2 Figs present the same results at the sequence level. Total reads are provided and are able to be compared because all the samples were analyzed on the same flow cell.

When comparing two samples, the two samples may have similar distributions of organisms in the microbial DNA. The abundance-based Jaccard index (Chao-Jaccard) was used to measure similarity with respect to the distribution of sequences [24, 25]. For each infection

A.

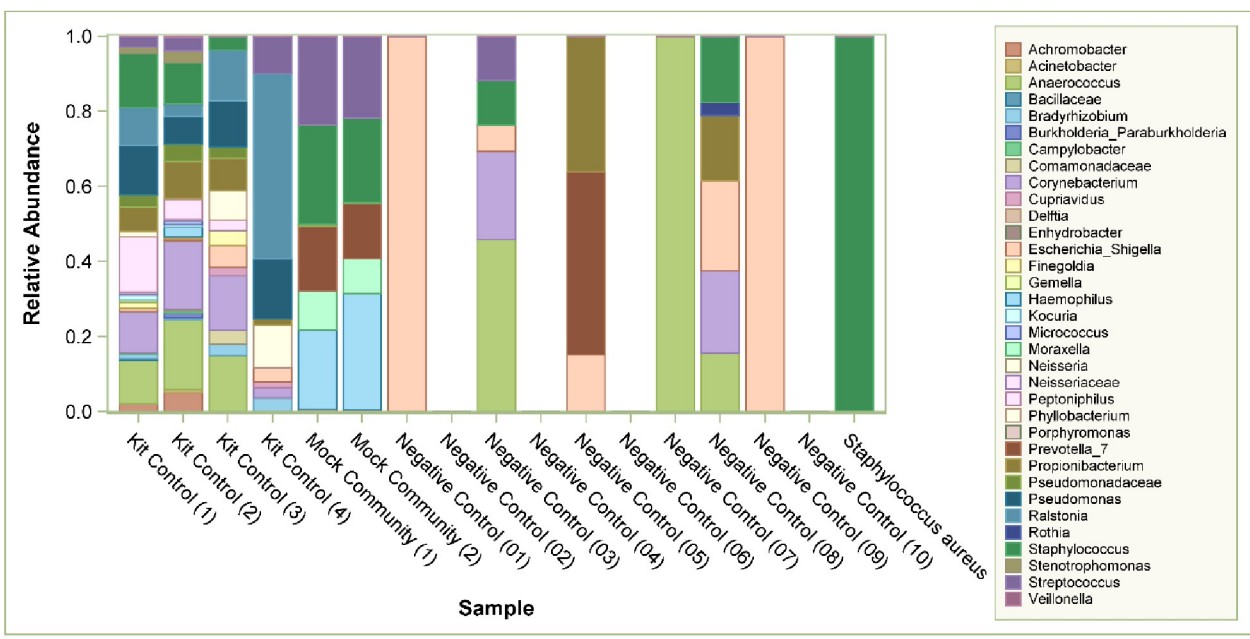

B.

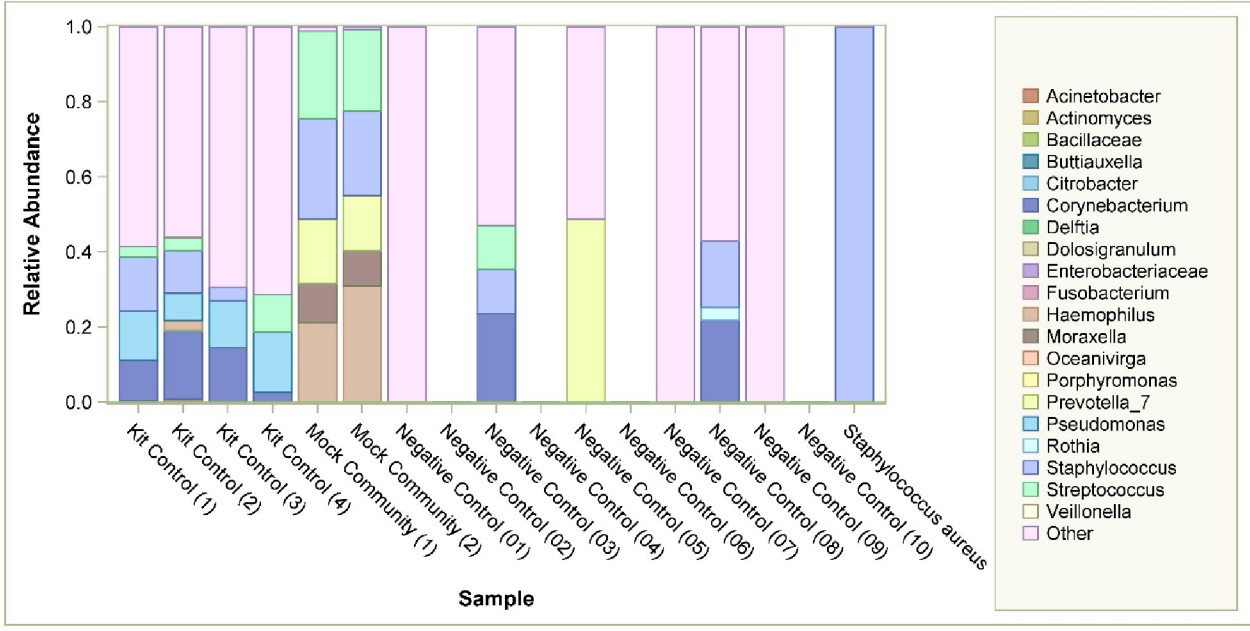

**Fig 1. High throughput sequencing for controls before (Panel A) and after Decontam (Panel B).** Relative abundance is demonstrated with bars with scale on left sided y-axis and legend provided. Control names are provided in text annotations along the x-axis.

dyad, sequence relative abundance from the last three CSF samples was compared to that from the first CSF sample from the subsequent infection. The Chao-Jaccard index of matched infection-reinfection pairs were compared with all other (unmatched) infection-reinfection pairs by permutation test to determine if the mean similarity measure of the paired samples is

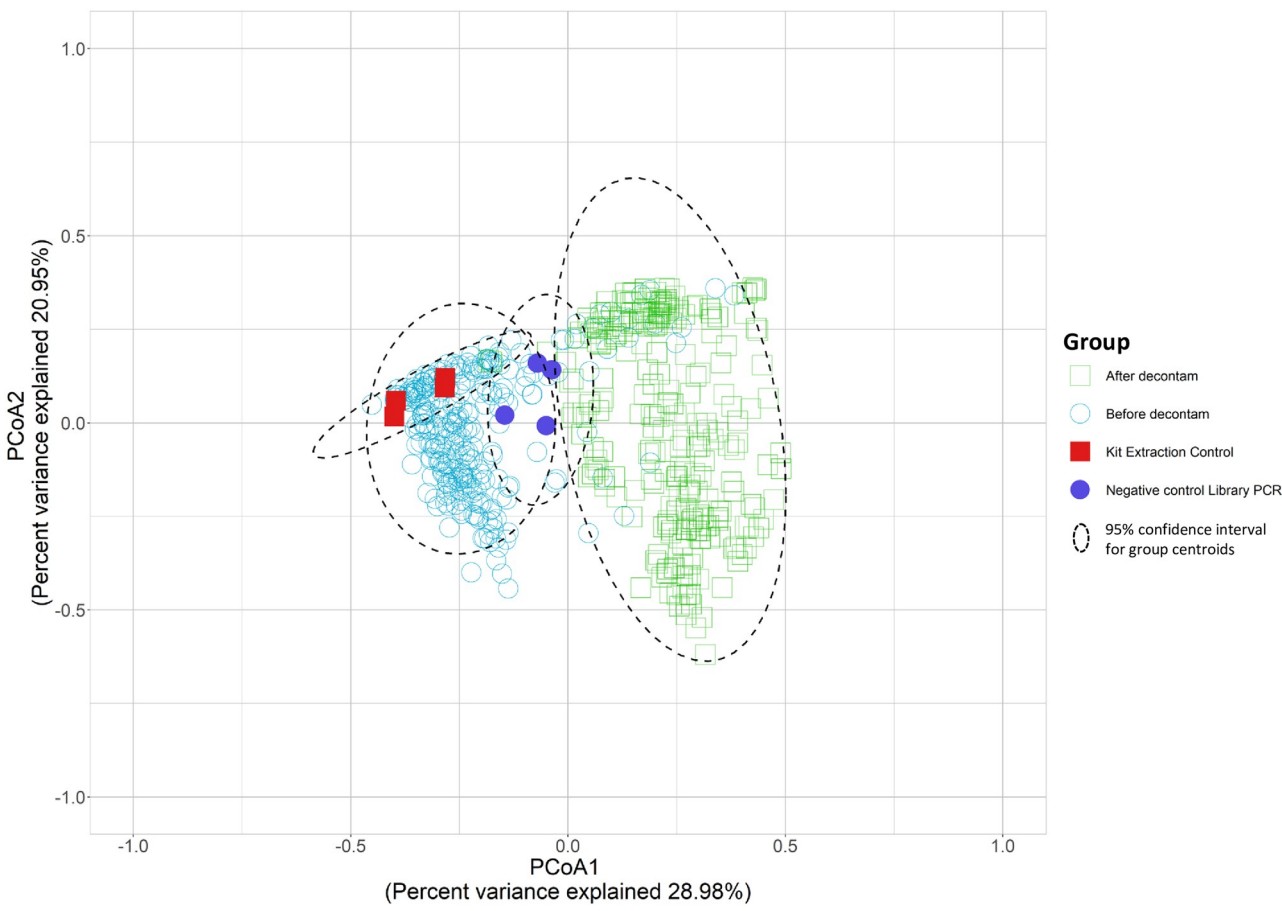

**Fig 2. Comparison of the CSF microbiota before and after decontam computational contaminant removal.** Principle coordinate analysis of the beta diversity (genus) values are based on Bray Curtis distances. Each color represents a microbiota grouping based on source. Ellipses were drawn using a confidence interval of 95% around the centroid for each group.

greater than the expected in samples with no association [26]. For the permutation tests, we calculated the similarity of randomly selected revision samples and randomly selected infection samples using Monte Carlo simulations. We generated a reference distribution of the mean similarity measures. The mean similarity from the matched pairs were compared to the reference distribution to determine significance.

Additional comparisons included the first culture of infection versus the aggregated last 3 cultures of the same infection; the first culture of the first infection versus first culture of the reinfection; and the association of Chao-Jaccard similarity measures with time between infections.

## Results

### Sequencing

Because the sequencing results from these CSF samples indicated that they had lower microbial loads than those found in many infection sample types, it was important to quantify the potential contribution of contaminating microbial DNA in the reagents used to process and sequence these samples. Therefore, as controls, we defined the microbiota in our DNA extraction kits ("Kit" in Table X, N = 4) and in the reagents used to prepare for sequencing [negative (i.e. no-template) controls, N = 10].

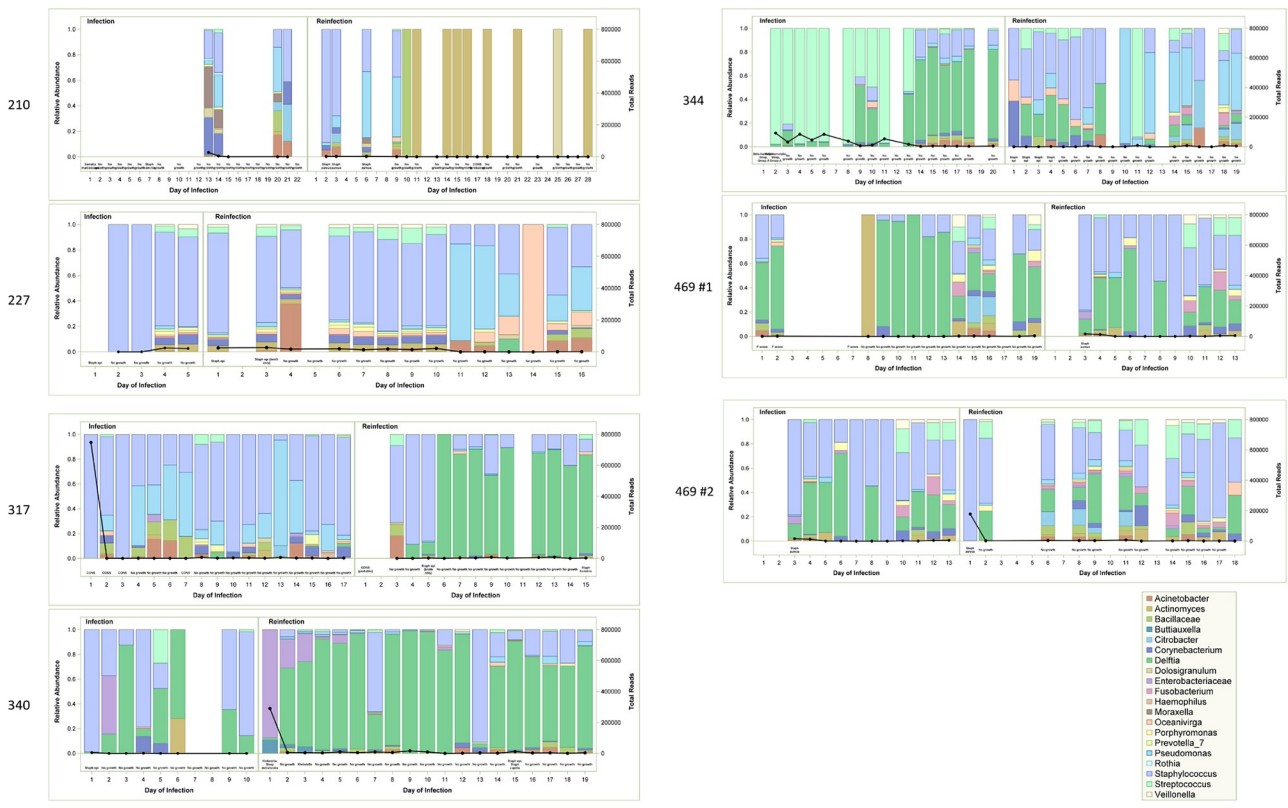

**Fig 3. High throughput sequencing results for sequential infections, probability-based contaminant removal (decontam).** Relative abundance is demonstrated with bars with scale on left-sided y-axis and legend provided. Total reads are demonstrated with heavy black line and with scale on right-sided y-axis. Bacterial culture results are provided in text annotations along the x-axis. While the analysis was conducted at sequence level, results are presented here at the genus level for ease of interpretability.

As shown in Table 2, the mean and median numbers of microbial reads detected in DNA extraction kit controls were higher than in the negative controls.

The aggregate genera from kit controls, duplicate mock community samples, 4 kit no-sample controls, 10 negative CSF controls, and a pure culture of *Staphylococcus aureus* are shown in Fig 1. Overall results (Fig 1A) differ from results where sequences identified as likely contaminants by the program decontam R were removed (Fig 1B).

While the median number of reads for both control types were lower than those for the CSF samples, there was some overlap in the ranges of CSF sample reads and those in controls, further highlighting the relatively low abundance of microbial DNA in the CSF infection samples. Therefore, we performed both computational analyses using decontam and full removal of contaminant sequences detected in extraction kit controls to adjust for the background microbiota in these samples.

We used principle coordinate analysis (PCoA) to investigate if the change in the communities after decontam was significant [27]. We found that the two highest-ranked dimensions, PCoA1 and PCoA2, explained 28.98% and 20.95% of variance respectively (Fig 2). The "After decontam" community composition significantly clustered away from the unfiltered "Before decontam" group following computational removal of contaminating reads by decontam (Adonis, P>0.001). Fig 2 also show members of the 'After decontam' group clustered apart from the Kit control group with no overlap at the 95% CI, indicating decontam removed

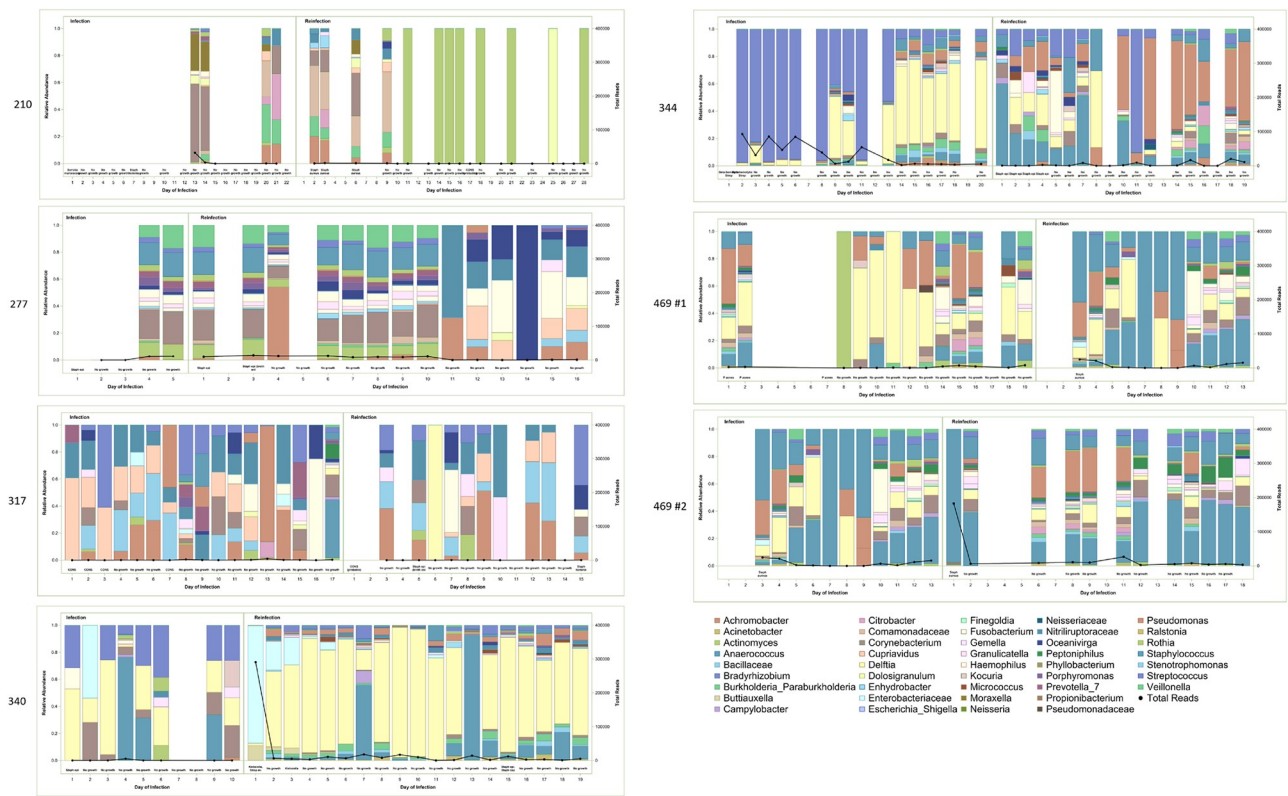

**Fig 4. High throughput sequencing results for sequential infections, removal of contaminant sequences detected in extraction kit controls.** Relative abundance is demonstrated with bars with scale on left-sided y-axis and legend provided. Total reads are demonstrated with heavy black line and with scale on right-sided y-axis. Bacterial culture results are provided in text annotations along the x-axis. While the analysis was conducted at sequence level, results are presented here at the genus level for ease of interpretability.

contaminating signal that originated from the DNA extraction kits. Some overlap remained between the 'After decontam" group and the negative library controls. However, it should be noted negative library control sequence reads were magnitudes lower than CSF sample reads, indicating the contribution of contaminating reads from library PCR reagents to CSF samples was likely minimal.

## CSF shunt infection results

We collected a total of 7 infection dyads for which CSF samples were available from the end of infection and beginning of reinfection; 5 patients had 2, and 1 patient had 3, sequential infections. An average of 13.5 samples per infection were analyzed. A total of 82 unique V4 sequences were recovered, of which 49 were identified as likely contaminants by the program decontam R and removed.

**Table 2. Summary of sequence reads by input sample type.**

| Input sample | Mean | Median | Range | n |
|---|---|---|---|---|
| CSF | 28,673 | 7,571 | 0–748,678 | 304 |
| Kit | 5,408 | 4089.5 | 809–12,563 | 4 |
| Negative control | 82.9 | 9 | 0–466 | 10 |

The proportion of bacterial species identified by HTS over the course of 7 CSF shunt infections in 6 different children are shown in Fig 3. Results are shown by infection dyad. For example, for patient 210 CSF culture on day 1 of initial infection identified *Serratia marcescens*; however, extra CSF for HTS was not available on that day. HTS was successfully performed on the last days of infection treatment, days 20 and 21; these identified a preponderance of *Staphylococcus* sequences. CSF culture from patient 210's reinfection recovered *Staphylococcus aureus* on days 2, 3, and 6; in addition, *Staphylococcus* sequences were most prevalent in the high throughput sequencing results on days 2, 3, and 6. Bacterial quantitation by qPCR detected $4 \times 10^5$ reads equivalents on day 13, followed by no or minimal bacterial DNA identified above the assay's limits of detection thereafter. Fig 3 presents results following probability-based contaminant removal (decontam). Similar results were observed following removal of contaminant sequences detected in extraction kit controls (Fig 4).

High throughput sequencing identified bacterial genera that were not observed in conventional culture. Sequencing of several of the samples identified a large variety of additional bacterial genera, each generally at relatively low abundance, and many of which are not considered typical CSF shunt infection pathogens.

Bacterial culture results are compared to high throughput sequencing findings in Table 3. Organisms detected by conventional culture were also identified at >44% relative abundances by high throughput sequencing in all but 1 infection (patient 469, infection 1) with CSF available. In 6 of 7 dyads, organisms identified by culture at reinfection were detected by high throughput sequencing of culture-negative samples at the end of the previous infection. A median of 8 sequences [interquartile range (IQR) 5–10] were observed in the first samples from reinfection.

CSF microbiota did change during over the course of infection treatment. (Table 4) The median Chao-Jaccard abundance-based similarity index from beginning to end of a single infection episode (intra-infection) was 0.67 (IQR 0.62–0.89) compared to that for unmatched infection pairs (inter-infection) of 0.41 (IQR 0.08–0.66) [p = 0.01]. Therefore, compared with separate infections, microbiota were significantly more similar within an infection episode.

Microbiota varied between infection and reinfection. (Table 4) The median Chao-Jaccard abundance-based similarity index for matched infection pairs at end of infection and beginning of reinfection using probability-based contaminant removal (decontam) was 0.57 (IQR 0.07–0.87) compared to that of unmatched pairs of 0.40 (IQR 0.10–0.60) [p = 0.46]. Following removal of contaminant sequences observed in extraction kit controls, median Chao-Jaccard abundance-based similarity index for matched infection pairs was 0.33 (IQR 0.25–0.53) compared to that of unmatched pairs of 0.17 (IQR 0.12–0.32) [p = 0.04]. Therefore, CSF microbiota at the end of infection and beginning of reinfection were more correlated than in any two unrelated infections after removal of all sequences observed in negative control samples. We reject the hypothesis that sequential infections were independent.

The mean time between infections (last available culture of infection to first available culture of reinfection) was 94 days (standard deviation 74 days), and ranged from 17 to 218 days. Time between infections was not significantly associated with abundance-based Chao-Jaccard similarity measures (r = -0.07; p = 0.88). Therefore, infection-reinfection dyads that were closer in time to one another were apparently not more similar than those separated by longer intervals.

## Discussion

To our knowledge, this is the first study that describes the microbiota present in CSF shunt infection and reinfection. Importantly, because the sequencing results from these CSF samples indicated that they had lower microbial loads than those found in many infection sample

**Table 3. Bacterial culture and high throughput sequencing (HTS) results (genus, read count, % of reads, and rank) and quantitative PCR (qPCR) for infection and reinfection dyads.**

| Patient | Infection | Infection Culture | Infection HTS | | Infection HTS | | qPCR |
|---|---|---|---|---|---|---|---|
| | Episode_Day | Organism | Genus | N | % | Rank | GE/ml of CSF |
| 210 | 1_1 | *Serratia marcescens* | N/A | N/A | N/A | N/A | Not done |
| 210 | 1_21 | No growth | *Staphylococcus* | 70 / 170 | 41% | 1 | Not done |
| 210 | 2_2 | *Staphylococcus aureus* | *Staphylococcus* | 1672 / 1883 | 89% | 1 | Not done |
| 227 | 1_1 | *Staphylococcus epidermidis* | N/A | N/A | N/A | N/A | Not done |
| 227 | 1_5 | No growth | *Staphylococcus* | 14968 / 21200 | 71% | 1 | Not done |
| 227 | 2_1 | *Staphylococcus epidermidis* | *Staphylococcus* | 19197 / 24494 | 78% | 1 | Undetected |
| 317 | 1_1 | *Staphylococcus epidermidis* | *Staphylococcus* | 747955 / 747955 | 100% | 1 | 1.87 x 10^6 |
| 317 | 1_17 | No growth | *Staphylococcus* | 1882 / 2386 | 79% | 1 | Undetected |
| 317 | 2_1 | *Staphylococcus epidermidis* | N/A | N/A | N/A | N/A | Not done |
| 317 | 2_3 | *Staphylococcus epidermidis* | *Staphylococcus* | 112 / 180 | 62% | 1 | Undetected |
| 340 | 1_1 | *Staphylococcus epidermidis* | *Staphylococcus* | 5091 / 5155 | 98% | 1 | Undetected |
| 340 | 1_10 | No growth | *Enterobacteriaceae Streptococcus* *Staphylococcus* | 0 / 180 3 / 180 151 / 180 | 0% 2% 84% | - 3 1 | Not done |
| 340 | 2_1 | *Klebsiella Streptococcus mitis/ oralis* | *Enterobacteriaceae Streptococcus* | 252875 / 290223 0 /290223 | 87% 0% | 1 - | 2.20 x 10^7 |
| 344 | 1_2 | Group A beta hemolytic *Streptococcus* | *Streptococcus* | 90562 / 92559 | 98% | 1 | Not done |
| 344 | 1_20 | No growth | *Staphylococcus Delftia* | 578 / 4514 3401 / 4514 | 13% 75% | 2 1 | Not done |
| 344 | 2_1 | *Staphylococcus epidermidis* | *Staphylococcus* | 69 / 158 | 44% | 1 | Undetected |
| 469 | 1_1 | *Propionibacterium acnes* | *Propionbacterium Delftia* | 0 / 845 397 / 845 | 0% 47% | - 1 | 7.57 x 10^6 |
| 469 | 1_19 | No growth | *Staphylococcus Delftia* | 641 / 3761 1596 / 3761 | 17% 42% | 2 1 | Undetected |
| 469 | 2_3 | *Staphylococcus aureus* | *Staphylococcus* | 12636 / 16125 | 78% | 1 | Not done |
| 469 | 2_13 | No growth | *Staphylococcus* | 2087 / 5089 | 41% | 1 | Undetected |
| 469 | 3_3 | *Staphylococcus aureus* | *Staphylococcus* | 177663 / 178450 | 99% | 1 | 1.05 x 10^6 |

N/A = sample not available for HTS; HTS N = read count; HTS % = percent reads; HTS Rank = rank of ordered relative abundance (1 = highest relative abundance).

types, it was critical to quantify the potential contribution of contaminating microbial DNA in the reagents used to process and sequence these CSF infection samples. The microbiota signal in the kit samples was critical to characterize and remove from subsequent analysis. Using high throughput sequencing and removing signal from reagents, we demonstrated that numerous bacterial taxa were detected in the CSF of children with both during CSF shunt infections and subsequent reinfections. These organisms were detected at high throughput sequencing counts well above those in negative controls. The organisms recovered by high throughput sequencing, including many taxa beyond those recovered by standard aerobic bacterial culture, suggest heterogeneous microbiota. However, the most abundant microbiota identified by high throughput sequencing results were generally consistent with those identified by culture in CSF shunt infection and reinfection. The composition of the microbiota changed over the course of infection treatment, likely in response to antibiotic use and surgical removal of the shunt but also potentially due to the introduction of new organisms through external ventricular drain(s) and/or externalized shunt apparatus. High throughput sequencing detected organisms missed by culture at the end of infection treatment but detected by

**Table 4. Similarity of microbiota over sequential infection episodes.**

| | Chao-Jaccard Abundance-Based Similarity Index | |
| --- | --- | --- |
| | **Median (IQR)** | |
| | Probability-based contaminant removal (decontam) | Removal of sequences observed in extraction kit controls |
| Sequential Infection (matched pairs) | | |
| Infection END to Reinfection BEGINNING | 0.57 (0.07, 0.87) | 0.33 (0.25, 0.53) |
| Infection BEGINNING to Reinfection BEGINNING | 0.40 (0.01, 0.82) | 0.18 (0.11, 0.57) |
| Infection BEGINNING to Infection END | 0.67 (0.62, 0.89) | 0.32 (0.04, 0.43) |
| Nonsequential Infection (unmatched pairs) | | |
| Infection END to Reinfection BEGINNING | 0.40 (0.10, 0.60) | 0.17 (0.12, 0.32) |
| Infection BEGINNING to Infection END | 0.41 (0.08, 0.66) | 0.13 (0.03, 0.22) |

culture at reinfection. However, these high throughput sequencing results also suggest that microbiota of consecutive infection episodes may be associated.

This work contributes to our understanding of the 'natural history' of infection treatment failure. Using longitudinal samples from children who have developed serial CSF shunt infections, our analysis demonstrated that CSF microbiota identified at the end of first infection are qualitatively similar to those seen at reinfection, and these infection/reinfection microbiota compositions may be correlated. Furthermore, infection-reinfection dyads that are closer in time to one another do not appear to be more similar than those separated by a longer time interval. Therefore, while these results indicate that molecular methods can characterize microbiota dynamics that extend beyond the organisms identified by culture during CSF infections, the clinical relevance of this changing microbiota composition is questionable.

The current findings suggest that microbes persist during antibiotic treatment. Since many devices such as catheters provide abiotic surfaces that could encourage or provide a substrate for biofilms that tend to be antibiotic resistant, future work will investigate whether biofilms play a role in these infections. Additional studies of cell pellets from the CSF, the shunt material and tubing itself, as well as studies of animal models and modes of infection, may shed light on the importance of biofilms for these findings.

There are several limitations to this work. The selection of a comprehensive set of appropriate negative controls is difficult. While laboratory-based controls such as kit controls, mock communities, and negative controls were always performed, optimal controls from uninfected human CSF were not available for this analysis. Future work will need to be performed to rigorously investigate whether sequencing detects microbiota in CSF from healthy children. We anticipate future work examining bacterial and fungal sequences will need to incorporate negative controls from several sources, including but not limited to: donor CSF, children with hydrocephalus undergoing shunt revision surgery for shunt failure, and children with hydrocephalus undergoing initial CSF shunt placement. In addition, future work will need to explore alternate sequencing approaches, including metagenomic sequencing.

In addition, there are several opportunities for contamination of CSF samples during recovery, storage, and experimentation [28]. However, sterile conditions were sought throughout recovery and storage of CSF samples, and contamination during experimentation is improbable given little or no detection of DNA in negative controls that were handled identically to and concurrently with the CSF samples. Therefore, contamination from PCR reagents is unlikely to have a major contribution to the bacterial diversity observed. A distinctive "kitome" (sequence reads derived from the DNA extraction kit reagents) was observed. The presence of the kitome may have the potential to affect the interpretation of CSF sample microbiota. In some cases the sample reads may become indistinguishable from the contaminating kitome reads. Decontam R effectively removed some kitome sequences, and similar results were observed following removal of contaminant sequences detected in extraction kit controls.

Testing from the hospital-certified laboratories was limited to routine aerobic cultures, practices that are subject to individual laboratory practices, including policies affecting identification of all organisms on a plate, incubation for extended periods of time, and enrichment for slow growing and fastidious organisms. We did not have information about variability or density of colonies on the original plates. In addition, anaerobic cultures were not obtained. Detection of microbial DNA, which could be from either viable or nonviable cells, by PCR does not guarantee that living bacteria or fungi were present at the time of sample collection. Alternately, high throughput sequencing may be detecting small colony variants and/or bacteria whose growth is being suppressed by antibiotics.

Finally, the small number of infection dyads analyzed limits further inquiry into factors associated with more or less similarity within dyads. For example, bimodality of similarity indices may suggest or distinguish between similar and dissimilar dyads. Such a distinction could further be used to assess risk factors associated with infection treatment failure or in the development of predictive models reflecting risk of infection by more harmful or difficult to treat bacteria.

## Conclusions

Despite its limitations, high throughput sequencing results were generally consistent with culture-based methods in CSF shunt infection and reinfection, and may detect organisms missed by culture at the end of infection treatment but detected by culture at reinfection. High throughput sequencing results are challenged by the low bacterial load present in CSF shunt infection, but suggest that microbiota of consecutive infection episodes may be associated.

## Supporting information

**S1 Fig. High throughput sequencing results for sequential infections, probability-based contaminant removal (decontam).** Relative abundance is demonstrated with bars with scale on left-sided y-axis and legend provided. Total reads are demonstrated with heavy black line and with scale on right-sided y-axis. Bacterial culture results are provided in text annotations along the x-axis. While the analysis was conducted at sequence level, results are presented here at the genus level for ease of interpretability.
(TIF)

**S2 Fig. High throughput sequencing results for sequential infections, removal of contaminant sequences detected in extraction kit controls.** Relative abundance is demonstrated with bars with scale on left-sided y-axis and legend provided. Total reads are demonstrated with heavy black line and with scale on right-sided y-axis. Bacterial culture results are provided in

text annotations along the x-axis. Results are presented here at the sequence level.
(TIF)

## Acknowledgments

Current membership of the CLIMB includes: Alexander Cheong, Gabriel Haller, Diego Morales, Alexander Rangel-Humphrey, Sabrina Sedano, and Lisa Wick. Past membership includes: Dan Berger, Whitney Bond, Haley Botteron, Courtney Dethlefs, Jessica Foster, Mohammed Gabir, Robert Johnson, Julie McGalliard, Deanna Mercer, Amanda Morgan, and Linda Shih.

We would like to thank the children and families who participated in the study at SCH and PCH. We thank those who made CSF sample collection at SCH and PCH possible, including: Anne J. Blaschke and Jay Riva-Cambrin who led enrollment at PCH; the pediatric neurosurgeons and neurosurgical staff at SCH and PCH; Hydrocephalus Clinical Research Network coordinators Amy Anderson and Tracey Habrock-Bach who identified eligible children to our study staff; and study staff who enrolled patients and ensured appropriate collection and storage of CSF specimens including Marshal Werfelman at SCH and Chris Stockmann (deceased), Priscilla Cowan, Trenda Barney, and Abby Phillips at PCH.

We also appreciate the assistance of the Hydrocephalus Clinical Research Network in providing clinical data from 2 participating centers.

## Author Contributions

**Conceptualization:** Tamara D. Simon.

**Data curation:** Christopher E. Pope, Jeffrey G. Ojemann.

**Formal analysis:** Kathryn B. Whitlock, Christopher E. Pope, Paul Hodor.

**Funding acquisition:** Tamara D. Simon.

**Investigation:** Christopher E. Pope, Lucas R. Hoffman, Tamara D. Simon.

**Methodology:** Kathryn B. Whitlock, Christopher E. Pope, Paul Hodor, Lucas R. Hoffman.

**Project administration:** Lucas R. Hoffman, David L. Limbrick, Jr., Patrick J. McDonald, Jason S. Hauptman, Tamara D. Simon.

**Resources:** Jeffrey G. Ojemann.

**Software:** Paul Hodor.

**Supervision:** Paul Hodor, Lucas R. Hoffman, David L. Limbrick, Jr., Patrick J. McDonald, Jason S. Hauptman, Jeffrey G. Ojemann, Tamara D. Simon.

**Visualization:** Paul Hodor.

**Writing – original draft:** Tamara D. Simon.

**Writing – review & editing:** Kathryn B. Whitlock, Christopher E. Pope, Paul Hodor, Lucas R. Hoffman, David L. Limbrick, Jr., Patrick J. McDonald, Jason S. Hauptman, Jeffrey G. Ojemann.

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
