## [Decision Letter · Decision Letter 0]

29 Apr 2020

PONE-D-19-35359

Characterization of cerebrospinal fluid (CSF) microbiota from patients with CSF shunt infection and reinfection using high throughput sequencing of 16S ribosomal RNA genes

PLOS ONE

Dear Dr. Simon,

Thank you for submitting your manuscript to PLOS ONE. After careful consideration, we feel that it has merit but does not fully meet PLOS ONE’s publication criteria as it currently stands. Therefore, we invite you to submit a revised version of the manuscript that addresses the points raised during the review process.

Both reviewers thought that the topic was significant, but both noted serious concerns that must be addressed adequately before further consideration can be made.

We would appreciate receiving your revised manuscript by Jun 13 2020 11:59PM. To enhance the reproducibility of your results, we recommend that if applicable you deposit your laboratory protocols in protocols.io, where a protocol can be assigned its own identifier (DOI) such that it can be cited independently in the future. For instructions see: http://journals.plos.org/plosone/s/submission-guidelines#loc-laboratory-protocols

We look forward to receiving your revised manuscript.

Kind regards,

Brenda A Wilson, Ph.D.

Academic Editor

PLOS ONE

Journal Requirements:

2. In the ethics statement in the Methods and online submission information, please ensure that you have specified what type of informed consent you obtained (for instance, written or verbal, and if verbal, how it was documented and witnessed).

Given that your study included minors, state whether you obtained consent from parents or guardians.

3. We note that you are reporting an analysis of a microarray, next-generation sequencing, or deep sequencing data set. PLOS requires that authors comply with field-specific standards for preparation, recording, and deposition of data in repositories appropriate to their field.

Please upload these data to a stable, public repository (such as ArrayExpress, Gene Expression Omnibus (GEO), DNA Data Bank of Japan (DDBJ), NCBI GenBank, NCBI Sequence Read Archive, or EMBL Nucleotide Sequence Database (ENA)). In your revised cover letter, please provide the relevant accession numbers for 16S rRNA sequencing data that may be used to access these data.

For a full list of recommended repositories, see http://journals.plos.org/plosone/s/data-availability#loc-omics or http://journals.plos.org/plosone/s/data-availability#loc-sequencing

5. Thank you for stating the following in the Financial Disclosure section:

'This study was supported by the National Institutes of Health via R01 NS095979 awarded to TDS, CEP, PH, LRH, DLL, PM. DLL receives research funding and equipment for unrelated projects through Microbot Medical, Inc. and Medtronic, Inc. LRH receives additional support from the National Institutes of Health via K24 HL141669.  The funders had no role in study design, data collection and analysis, decision to publish, or preparation of the manuscript.'

We note that you received funding from a commercial source: Microbot Medical, Inc. and Medtronic, Inc

6. We note you have included a table to which you do not refer in the text of your manuscript. Please ensure that you refer to Table 1 in your text; if accepted, production will need this reference to link the reader to the Table.

7. Please upload a copy of Supporting Information Table S1 which you refer to in your text on page 9.

Reviewers' comments:

Reviewer's Responses to Questions

**Comments to the Author**

1. Is the manuscript technically sound, and do the data support the conclusions?

Reviewer #1: No

Reviewer #2: Partly

2. Has the statistical analysis been performed appropriately and rigorously? 

Reviewer #1: No

Reviewer #2: No

3. Have the authors made all data underlying the findings in their manuscript fully available?

Reviewer #1: No

Reviewer #2: No

4. Is the manuscript presented in an intelligible fashion and written in standard English?

Reviewer #1: Yes

Reviewer #2: Yes

5. Review Comments to the Author

Reviewer #1: The study conducted by Whitlock et al. describes the microbiota of CSF shunt infections and recurrent infections. The main objective is to test whether taxa undetected by conventional culture techniques at the end of the first infection, but detected in subsequent infections, can be detected by sequencing at both time points (taking advantage of the resolution of highthroughput sequencing). The overall objective of the study is interesting, and the manuscript is in general well written. The authors have used proper controls were possible (extraction kit control, PCR and sequencing control). However, I have main concerns regarding the existence of contaminant sequences in the dataset even after authors’ attempt to remove these contaminants using a popular bioinformatics package (decontam). CSF is considered an sterile biological fluid, and even in case of infection, authors should have expected very low bacterial load in samples. Reagent contaminations could have been minimized prior to extraction and sequencing. A more stringent removal of contaminant reads is required to properly interpret the results. Also, bioinformatics analyses should focus on taking advantage of the information of sequence variants rather than using bacterial genus level.

Please see my specific comments below:

Abstract:

Lines 33-34: not clear how many samples from each patient? 2 in total, or at the end of each (re)infection and begging of next reinfection? This needs to be clarified in the methods section as well.

Lines 38-39: need to briefly clarify the method by which 8 taxa were detected? HTS or culture? Are these unique amplicon sequence variants (ASVs)?

Introduction:

Line 71: “over 2,000 CSF shunt infections diagnosed each year” worldwide or the US?

In the introduction, please provide information regarding pathogens/opportunistic commonly isolated from CSF shunt infections (if any literature available).

Methods:

Lines 115-123: Still not clear how many samples were collected per patient? In line 105, authors indicate “we examined serial CSF obtained from the subset of children”, but never refer to exact numbers until lines 243-245. Please clarify this in the “specimen collection” section.

Lines 129-134: would be informative to provide a one line description of the conventional culture (media used, incubation time and temperature).

Results:

Lines 272-273: Color codes used in Figure1 are a bit difficult to differentiate, but apparently even after removal of the kit contaminant sequences, reads belonging to important taxa such as Staphylococcus, Streptococcus, Pseudomonas, and Corynebacterium remain in the dataset. Based on Figure3, these bacterial taxa are abundant in the final microbiota profile of biological samples. There are main limitations here:

- Taxa indicated in the bar charts, are these based on the proportion of bacterial genera (genus level) or ASV? If based on genera, the results are simply not acceptable. You are losing resolution, you could have streptococcal ASVs in the kit contaminant with different sequence than those detected in biological samples. Combing information at genus level is misleading. IF the information provided is based on genus level, please reanalyze the data and present it at the ASV level (both with respect to bar charts and PCoA analysis).

- The decontam package was not developed to (and is not able to) completely fix the issue of laboratory reagent contamination, particularly when it comes to low biomass samples. Most of the taxa detected in reagents are successful environmental survivors capable of producing biofilms. Members of these taxa are also omnipresent in hospitals, can be potential pathogens causing human diseases, or even found as opportunistic inside or on different surfaces of our body (e.g. skin). With respect to CFS, which is considered an sterile biological fluid, one should expect zero to very low levels of bacterial DNA even in the case of CSF shunt infections. The proper way to address this issue was to pre-treat extraction kit and PCR reagents with DNAse prior to conducting the experiment (there are even commercial kits available for this purpose). Now that this method has been neglected, the most proper way to analyze a sensitive dataset like CSF is to completely remove contaminant ASVs detected in extraction kit controls from the biological samples. Then report the total number of reads and microbiota profile before and after removal of contaminant ASVs (this can be done and presented in parallel to the of decontam to enable unbiased comparison). Of course, this approach could lead to removal of biologically relevant ASVs originating from samples, but this is the only way to make the results reliable.

- A concerning point in the post-decontam microbiota profile of biological samples is the presence of several bacterial groups that should have been detected by aerobic culture (most members of staphylococcus, pseudomonas, coryebacterium, etc. are not fastidious and grow aerobically). The absence of these taxa from culture results is additional proof for possibility of contamination.

- Next limitation of decontam package or even complete removal of contaminant DNAs is between-kit variation in the profile of contaminant DNA. In Figure1A (and less in Figure1B) it is evident that each batch of kit has it’s own microbiota profile, making removal of contaminant reads more challenging. So, when re-analyzing the dataset by complete removal of contaminant DNA, each biological sample should be filtered based on its corresponding extraction kit profile.

Lines 287-289: “indicating decontam removed contaminating signal that originated from the DNA extraction kits”. Not necessarily, significant clustering pattern does not equal to removal of contaminant DNA. As indicated, you have shared contaminant taxa (and most likely ASVs) in your extraction kit control and biological samples even after using decontam.

Lines 294-297: Why doing beta-diversity analysis at the genus level? You have used dada2 algorithm and have access to ASVs, please use that data.

Figure2: Also indicate on the PCoA graph the kit control and negative controls after the decontam package.

Line 301-303: Again, please provide information whether the streptococcal, staylococcal, pseudomonas, and Corynebacterial ASVs remaining in the kit extraction control after decontam are the same/different from the ones detected in post-decontam microbiota profile of the biological samples.

Lines 311-314: Regardless of whether performed on the same flowcell or not, sequencing reads cannot be used for quantification. There are many technological limitations to this, please revise this section.

Discussion:

The authors have identified and acknowledged potential limitations of the study. However, as indicated earlier, most of the results/discussion of this manuscript is based on the ability of decontam package to completely remove contaminant reads, which is not a true statement. Particularly, the authors need to expand this limitation by discussing why standard culture failed to recover supposedly aerobic taxa detected by sequencing.

Data availability:

The sequencing reads and metadata (sample information) should become available to allow careful examination of the results.

Reviewer #2: This report describes a study aimed at addressing an important biomedical problem, namely whether reinfections in cerebrospinal fluid (CSF) infections are caused by microbes previously present or are from independent new introductions. The authors examined CSF samples taken from end of infection and beginning of reinfection from 6 patients, each with 2-3 infections. The V4 regions of the 16S rRNA genes were sequenced and the taxonomic assignments compared to those obtained from bacterial species found in cultures from the samples. The results from direct sequencing of samples were consistent with that obtained from the bacterial cultures, but the CSF microbial profiles did not correlate between end of infection and beginning of reinfection. Overall the manuscript is fairly well written. It is commendable that the authors considered the possibility that there might be contamination from reagents and tried to remove artifactual data bioinformatically. The authors have also acknowledged many of the limitations of their study and analysis approach. However, there are a number of concerns with the general experimental setup and data interpretation that need to be addressed.

1. It is not clear why only the V4 region of 16S rRNA gene was sequenced and why metagenomic sequencing was not performed.

2. It is very difficult to assess the importance of the relative abundance of the microbial contents of the sample for which the total microbial amount is uncertain. The microbial burden of each sample should be more reliably and accurately determined. Sequencing reads are inappropriate for quantification purposes. And, relative abundance does not speak to total abundance.

3. Use of the Decontam program does not convincingly remove all contaminating sequences such that a good quality representation of the actual sample is obtained. It is surprising that sequences for aerobic, relatively hardy taxa were found in the 16S dataset but not detected in the cultured dataset. This strongly suggests the possibility of the results coming from contamination. This concern is further supported by the results shown in Figure 1, where each kit batch has a different profile.

4. It appears that the figures are showing results of taxa based on genus level not ASV. Why was this done? ASV is higher resolution and should be able to distinguish contaminant reads from sample reads.

5. The description of the number of samples collected and analyzed per patient and which ones exactly were used for analysis is not clear. After much thought it is possible to figure it out, but more clarity would be helpful.

6. For the discussion, it would be helpful to provide more comparative information of the results from the current study with the types of microbes frequently found associated with CSF infections/reinfections, based on the literature. Since there are over 2000 cases every year (assuming in US??), one would think that there would be information regarding the types of microbes associated with these infections. Are there aerobic bacteria often found?

6. PLOS authors have the option to publish the peer review history of their article (what does this mean?). If published, this will include your full peer review and any attached files.

Reviewer #1: No

Reviewer #2: No

---

## [Author Response · Author response to Decision Letter 0]

8 Dec 2020

October 26, 2020

Dear PLOS One Editorial team,

We appreciate the consideration given by your reviewers and the editorial board to our original research article PONE-D-19-35359 entitled “Characterization of cerebrospinal fluid (CSF) microbiota from patients with CSF shunt infection and reinfection using high throughput sequencing of 16S ribosomal RNA genes”. 

We are responding to the critiques provided with the following changes (original feedback is provided in italics): 

1. Journal requirements: Please ensure that your manuscript meets PLOS ONE's style requirements, including those for file naming. 

Figure files and supporting information were reformatted and renamed to comply with guidelines. In addition, edits were made to the title page to comply with PLOS One formatting requirements.

2. Journal requirements: In the ethics statement in the Methods and online submission information, please ensure that you have specified what type of informed consent you obtained (for instance, written or verbal, and if verbal, how it was documented and witnessed). Given that your study included minors, state whether you obtained consent from parents or guardians.

We have added clarification that “written consent was obtained from parents or guardians, and assent when age- and developmentally-appropriate from study subjects” to the ethics statement.

3. Journal requirements: We note that you are reporting an analysis of a microarray, next-generation sequencing, or deep sequencing data set. PLOS requires that authors comply with field-specific standards for preparation, recording, and deposition of data in repositories appropriate to their field.

We were able to upload our data to NCBI Sequence Read Archive PRJNA608523 on February 24, 2020 with a release date in June 2020. 

4. Journal requirements: We note that you have stated that you will provide repository information for your data at acceptance. Should your manuscript be accepted for publication, we will hold it until you provide the relevant accession numbers or DOIs necessary to access your data. If you wish to make changes to your Data Availability statement, please describe these changes in your cover letter and we will update your Data Availability statement to reflect the information you provide.

We were able to upload our data to NCBI Sequence Read Archive PRJNA608523 on February 24, 2020 with a release date in June 2020. We have revised the cover letter accordingly. 

5. Journal requirements: Thank you for stating the following in the Financial Disclosure section:

'This study was supported by the National Institutes of Health via R01 NS095979 awarded to TDS, CEP, PH, LRH, DLL, PM. DLL receives research funding and equipment for unrelated projects through Microbot Medical, Inc. and Medtronic, Inc. LRH receives additional support from the National Institutes of Health via K24 HL141669. The funders had no role in study design, data collection and analysis, decision to publish, or preparation of the manuscript.' We note that you received funding from a commercial source: Microbot Medical, Inc. and Medtronic, Inc

Within this Competing Interests Statement, please confirm that this does not alter your adherence to all PLOS ONE policies on sharing data and materials by including the following statement: "This does not alter our adherence to PLOS ONE policies on sharing data and materials.” 

We have included the two commercial sources above in Dr. Limbrick’s funding disclosures on the PLOSOne website. We have provided the financial disclosure statement above in the cover letter, and provided the additional statement according to the above guidance. 

 We have included the two commercial sources above in Dr. Limbrick’s funding disclosures on the PLOSOne website. We have provided the financial disclosure statement above in the cover letter, and provided the additional statement according to the above guidance. We appreciate the additional support.

6. We note you have included a table to which you do not refer in the text of your manuscript. Please ensure that you refer to Table 1 in your text; if accepted, production will need this reference to link the reader to the Table.

 We have revised an incorrect reference to Table S1 to a correct reference to Table 1. 

7. Please upload a copy of Supporting Information Table S1 which you refer to in your text on page 9.

 We have revised an incorrect reference to Table S1 to a correct reference to Table 1. 

8. Reviewer #1, general comments: The authors have used proper controls were possible (extraction kit control, PCR and sequencing control). However, I have main concerns regarding the existence of contaminant sequences in the dataset even after authors’ attempt to remove these contaminants using a popular bioinformatics package (decontam). CSF is considered an sterile biological fluid, and even in case of infection, authors should have expected very low bacterial load in samples. Reagent contaminations could have been minimized prior to extraction and sequencing. A more stringent removal of contaminant reads is required to properly interpret the results. 

 We appreciate this feedback and have provided additional analyses using the more stringent approach suggested. We have revised the methods section accordingly, provided new results in a new Figure 4 and the original Table 4. We did observe a statistically significant association using this approach, and hence have revised the results and discussion accordingly. Please see specific comments below for additional details about revisions made.

9. Reviewer #1, general comments: Also, bioinformatics analyses should focus on taking advantage of the information of sequence variants rather than using bacterial genus level.

We appreciate this feedback. The original analysis was conducted at the sequence level, and original figures presented at genus level for interpretability. We have now provided results at the sequence level in the Supplemental Figures. We have also provided this clarification in revisions to the methods section as well as figure legends. Please see specific comments below for additional details about revisions made. 

10. Reviewer #1, specific comments: Abstract: Lines 33-34: not clear how many samples from each patient? 2 in total, or at the end of each (re)infection and begging of next reinfection? This needs to be clarified in the methods section as well.

We have revised the abstract methods to read, “Serial CSF samples were obtained from 6 patients, 5 with 2 infections and 1 with 3 infections; the study was limited to those for which CSF samples were available from the end of infection and beginning of reinfection.”

We included the following statement in the Study Subjects section of the Methods as well: “We analyzed serial CSF samples from 6 patients, 5 with 2 infections and 1 with 3 infections; the study was limited to those for which CSF samples were available from the end of infection and beginning of reinfection.”

11. Reviewer #1, specific comments: Abstract Lines 38-39: need to briefly clarify the method by which 8 taxa were detected? HTS or culture? Are these unique amplicon sequence variants (ASVs)?

 We have now clarified that 8 taxa were identified using HTS. These were analyzed at a sequence level; figures were originally presented at genus level for interpretability. The original analysis was conducted at the sequence level, and original figures presented at genus level for interpretability. We have now provided results at the sequence level in the Supplemental Figures.

12. Reviewer #1, specific comments: Introduction: Line 71: “over 2,000 CSF shunt infections diagnosed each year” worldwide or the US?

 We have now clarified that there are over 2,000 shunt infections annually in the U.S.

13. Reviewer #1, specific comments: In the introduction, please provide information regarding pathogens/opportunistic commonly isolated from CSF shunt infections (if any literature available).

Thank you. We have added a statement: “The most common pathogens recovered from conventional microbiologic cultures include primarily Staphylococcus epidermidis and Staphylococcus aureus, followed by gram-negative organisms.[ref]”

14. Reviewer #1, specific comments: Methods: Lines 115-123: Still not clear how many samples were collected per patient? In line 105, authors indicate “we examined serial CSF obtained from the subset of children”, but never refer to exact numbers until lines 243-245. Please clarify this in the “specimen collection” section.

We included the following statement: “We analyzed serial CSF samples from 6 patients, 5 with 2 infections and 1 with 3 infections; the study was limited to those for which CSF samples were available from the end of infection and beginning of reinfection.” We elected to update the Study Subjects section rather than the Specimen Collection section of the Methods.

15. Reviewer #1, specific comments: Lines 129-134: would be informative to provide a one line description of the conventional culture (media used, incubation time and temperature).

Because the media and incubation approaches can differ based on organism recovered, we have elected to retain the original description: “Conventional cultures are the traditional diagnostic approach used to detect typical pathogens in infectious diseases and were performed in a clinical microbiology laboratory following Clinical and Laboratory Standards Institute guidelines”. We had included in the limitation section the following statement as well: “Testing from the hospital-certified laboratories was limited to routine aerobic cultures, practices that are subject to individual laboratory practices, including policies affecting identification of all organisms on a plate, incubation for extended periods of time, and enrichment for slow growing and fastidious organisms. We did not have information about variability or density of colonies on the original plates.”

16. Reviewer #1, specific comments: Results: Lines 272-273: Color codes used in Figure1 are a bit difficult to differentiate, but apparently even after removal of the kit contaminant sequences, reads belonging to important taxa such as Staphylococcus, Streptococcus, Pseudomonas, and Corynebacterium remain in the dataset. Based on Figure 3, these bacterial taxa are abundant in the final microbiota profile of biological samples. There are main limitations here:

- Taxa indicated in the bar charts, are these based on the proportion of bacterial genera (genus level) or ASV? If based on genera, the results are simply not acceptable. You are losing resolution, you could have streptococcal ASVs in the kit contaminant with different sequence than those detected in biological samples. Combing information at genus level is misleading. IF the information provided is based on genus level, please reanalyze the data and present it at the ASV level (both with respect to bar charts and PCoA analysis).

We appreciate that we need to be far more explicit about our approach. We have added the following statement to the Data Analysis section: “While the analysis was conducted at sequence level, Figures 1, 3, and 4 present results at the genus level for ease of interpretability.”

We have also added results at the sequence level and go on to say, “Supplemental Figures 1 and 2 present the same results at the sequence level.”

We have edited the legends to Figures 1, 3, and 4.

We have also revised the limitation section of the discussion to read, “Decontam R effectively removed some kitome sequences, and similar results were observed following removal of contaminant sequences detected in extraction kit controls.”

17. Reviewer #1, specific comments: Results: - The decontam package was not developed to (and is not able to) completely fix the issue of laboratory reagent contamination, particularly when it comes to low biomass samples. Most of the taxa detected in reagents are successful environmental survivors capable of producing biofilms. Members of these taxa are also omnipresent in hospitals, can be potential pathogens causing human diseases, or even found as opportunistic inside or on different surfaces of our body (e.g. skin). With respect to CFS, which is considered an sterile biological fluid, one should expect zero to very low levels of bacterial DNA even in the case of CSF shunt infections. The proper way to address this issue was to pre-treat extraction kit and PCR reagents with DNAse prior to conducting the experiment (there are even commercial kits available for this purpose). Now that this method has been neglected, the most proper way to analyze a sensitive dataset like CSF is to completely remove contaminant ASVs detected in extraction kit controls from the biological samples. Then report the total number of reads and microbiota profile before and after removal of contaminant ASVs (this can be done and presented in parallel to the of decontam to enable unbiased comparison). Of course, this approach could lead to removal of biologically relevant ASVs originating from samples, but this is the only way to make the results reliable.

Thank you for the suggestion. We have now provided additional analyses using the more stringent approach suggested. We have revised the methods section to explain, “Two strategies were used to address contaminant sequences. … Alternately, sequences detected in extraction kit controls were identified as contaminants and were removed from the CSF samples.” 

 We have provided results in a new Figure 4. The results section now reads, “Figure 3 presents results following probability-based contaminant removal (decontam). Similar results were observed following removal of contaminant sequences detected in extraction kit controls (Figure 4).”

 We have also updated the Chao-Jaccard abundance-based similarity index in Table 4. The associated text in the Results section now reads, “Following removal of contaminant sequences observed in extraction kit controls, median Chao-Jaccard abundance-based similarity index for matched infection pairs was 0.33 (IQR 0.25-0.53) compared to that of unmatched pairs of 0.17 (IQR 0.12-0.32) [p = 0.04].”

 We did observe a statistically significant association using this approach, and hence have revised the results section accordingly: “Therefore, CSF microbiota at the end of infection and beginning of reinfection were more correlated than in any two unrelated infections after removal of all sequences observed in negative control samples. We reject the hypothesis that sequential infections were independent.”

 The discussion and conclusion have been revised to state that the high throughput sequencing results also suggest that microbiota of consecutive infection episodes may be associated.

18. Reviewer #1, specific comments: Results: - A concerning point in the post-decontam microbiota profile of biological samples is the presence of several bacterial groups that should have been detected by aerobic culture (most members of staphylococcus, pseudomonas, coryebacterium, etc. are not fastidious and grow aerobically). The absence of these taxa from culture results is additional proof for possibility of contamination.

 We appreciate this feedback and have provided additional analyses using the more stringent approach suggested. As already described, we have revised the methods section accordingly, provided new results in a new Figure 4 and the original Table 4. We did observe a statistically significant association using this approach, and hence have revised the results and discussion accordingly. 

 We also would suggest that high throughput sequencing detects bacterial DNA rather than culturable cells. We had previously stated in the discussion section, “Detection of microbial DNA, which could be from either viable or nonviable cells, by PCR does not guarantee that living bacteria or fungi were present at the time of sample collection.” We have now added the statement, “Alternately, high throughput sequencing may be detecting small colony variants and/or bacteria whose growth is being suppressed by antibiotics.” 

19. Reviewer #1, specific comments: Results: - Next limitation of decontam package or even complete removal of contaminant DNAs is between-kit variation in the profile of contaminant DNA. In Figure1A (and less in Figure1B) it is evident that each batch of kit has it’s own microbiota profile, making removal of contaminant reads more challenging. So, when re-analyzing the dataset by complete removal of contaminant DNA, each biological sample should be filtered based on its corresponding extraction kit profile.

 We appreciate this feedback and have provided additional analyses exactly as suggested. 

20. Reviewer #1, specific comments: Results: Lines 287-289: “indicating decontam removed contaminating signal that originated from the DNA extraction kits”. Not necessarily, significant clustering pattern does not equal to removal of contaminant DNA. As indicated, you have shared contaminant taxa (and most likely ASVs) in your extraction kit control and biological samples even after using decontam.

 We appreciate this feedback and have provided additional analyses using the more stringent approach suggested. As already described, we have revised the methods section accordingly, provided new results in a new Figure 4 and the original Table 4. We did observe a statistically significant association using this approach, and hence have revised the results and discussion accordingly. 

21. Reviewer #1, specific comments: Results: Lines 294-297: Why doing beta-diversity analysis at the genus level? You have used dada2 algorithm and have access to ASVs, please use that data.

We appreciate this feedback. The original analysis was conducted at the sequence level, and original figures presented at genus level for interpretability. We have now provided results at the sequence level in the Supplemental Figures. We have also provided this clarification in revisions to the methods section as well as figure legends. Please see specific comments above for additional details about revisions made.

22. Reviewer #1, specific comments: Results: Figure2: Also indicate on the PCoA graph the kit control and negative controls after the decontam package.

We appreciate this suggestion, and affirm that Figure 2 provides the reader feedback on the effectiveness of decontam for contaminant removal. Because we have performed the additional analysis with complete removal of contaminant sequences, we believe this suggestion is less critical and have elected to defer.

23. Reviewer #1, specific comments: Results: Line 301-303: Again, please provide information whether the streptococcal, staylococcal, pseudomonas, and Corynebacterial ASVs remaining in the kit extraction control after decontam are the same/different from the ones detected in post-decontam microbiota profile of the biological samples.

Thank you for this question. While decontam removed many sequences that were apparent in extraction kit control samples, some of the sequences present in extraction kit controls were also detected in the post-decontam microbiota profile of the biological samples. Of 10 Corynebacterial ASVs, 5 were removed via decontam. Of the remaining sequences, 4 were not present in any extraction kit control sample; a single sequence was present in extraction kit 2 only. Of 8 Streptococcal ASVs, 5 were removed via decontam. Of the remaining sequences, 2 were not present in any extraction kit control sample; a single sequence was present in extraction kit 1 only. Of 3 Staphylococcal ASVs, none was removed by decontam. One of the sequences was not present in any extraction kit controls; 1 was present in kit 2 only; and 1 was present in kits 1, 2, and 3. Of 8 Pseudomonas ASVs, 4 were removed via decontam. Of the remaining sequences, 3 did not appears in any extraction kit control sample; a single sequence was present in extraction kits 1 and 2. 

24. Reviewer #1, specific comments: Results: Lines 311-314: Regardless of whether performed on the same flowcell or not, sequencing reads cannot be used for quantification. There are many technological limitations to this, please revise this section.

 We appreciate the reviewers point and have removed this statement.

25. Reviewer #1, specific comments: Discussion: The authors have identified and acknowledged potential limitations of the study. However, as indicated earlier, most of the results/discussion of this manuscript is based on the ability of decontam package to completely remove contaminant reads, which is not a true statement. Particularly, the authors need to expand this limitation by discussing why standard culture failed to recover supposedly aerobic taxa detected by sequencing.

 The reviewer raises an excellent point. We have revised the limitations to reflect the additional analysis, and it now reads, “Decontam R effectively removed some kitome sequences, and similar results were observed following removal of contaminant sequences detected in extraction kit controls.”

We also would suggest that high throughput sequencing detects bacterial DNA rather than culturable cells. We had previously stated in the discussion section, “Detection of microbial DNA, which could be from either viable or nonviable cells, by PCR does not guarantee that living bacteria or fungi were present at the time of sample collection.” We have now added the statement, “Alternately, high throughput sequencing may be detecting small colony variants and/or bacteria whose growth is being suppressed by antibiotics.”

26. Reviewer #1, specific comments: Data availability: The sequencing reads and metadata (sample information) should become available to allow careful examination of the results.

We were able to upload our data to NCBI Sequence Read Archive PRJNA608523 on February 24, 2020 with a release date in June 2020.

27. Reviewer #2: 1. It is not clear why only the V4 region of 16S rRNA gene was sequenced and why metagenomic sequencing was not performed.

We chose the 16S region because it covers bacterial taxa cultured from CSF and there are a substantial number of human sequences present. We are exploring metagenomics as a new direction, and have added the following statement to the discussion: “In addition, future work will need to explore alternate sequencing approaches, including metagenomic sequencing.”. 

28. Reviewer #2: 2. It is very difficult to assess the importance of the relative abundance of the microbial contents of the sample for which the total microbial amount is uncertain. The microbial burden of each sample should be more reliably and accurately determined. Sequencing reads are inappropriate for quantification purposes. And, relative abundance does not speak to total abundance.

We acknowledge the limitations of total reads, and hence present them beside standard quantification using quantitative 16S PCR results in Table 3. As shown, quantitative PCR is rarely detected in CSF shunt infection, so total reads are presented to provide the reader additional indirect evidence to the low bacterial load present. We have also removed the statement in reference to figure 3 that read, “Because all the samples were analyzed on the same flow cell, total reads are able to be compared.” 

29. Reviewer #2: 3. Use of the Decontam program does not convincingly remove all contaminating sequences such that a good quality representation of the actual sample is obtained. It is surprising that sequences for aerobic, relatively hardy taxa were found in the 16S dataset but not detected in the cultured dataset. This strongly suggests the possibility of the results coming from contamination. This concern is further supported by the results shown in Figure 1, where each kit batch has a different profile.

Thank you for the suggestion. We have now provided additional analyses using a more stringent approach. We have revised the methods section to explain, “Two strategies were used to address contaminant sequences. … Alternately, sequences detected in extraction kit controls were identified as contaminants and were removed from the CSF samples.” 

 We have provided results in a new Figure 4. The results section now reads, “Figure 3 presents results following probability-based contaminant removal (decontam). Similar results were observed following removal of contaminant sequences detected in extraction kit controls (Figure 4).”

 We have also updated the Chao-Jaccard abundance-based similarity index in Table 4. The associated text in the Results section now reads, “Following removal of contaminant sequences observed in extraction kit controls, median Chao-Jaccard abundance-based similarity index for matched infection pairs was 0.33 (IQR 0.25-0.53) compared to that of unmatched pairs of 0.17 (IQR 0.12-0.32) [p = 0.04].”

 We did observe a statistically significant association using this approach, and hence have revised the results section accordingly: “Therefore, CSF microbiota at the end of infection and beginning of reinfection were more correlated than in any two unrelated infections after removal of all sequences observed in negative control samples. We reject the hypothesis that sequential infections were independent.”

 The discussion and conclusion have been revised to state that the high throughput sequencing results also suggest that microbiota of consecutive infection episodes may be associated.

30. Reviewer #2: 4. It appears that the figures are showing results of taxa based on genus level not ASV. Why was this done? ASV is higher resolution and should be able to distinguish contaminant reads from sample reads.

We appreciate that we need to be far more explicit about our approach. We have added the following statement to the Data Analysis section: “While the analysis was conducted at sequence level, Figures 1, 3, and 4 present results at the genus level for ease of interpretability.”

We have also added results at the sequence level and go on to say, “Supplemental Figures 1 and 2 present the same results at the sequence level.”

We have similarly edited the legends to Figures 1, 3, and 4.

31. Reviewer #2: 5. The description of the number of samples collected and analyzed per patient and which ones exactly were used for analysis is not clear. After much thought it is possible to figure it out, but more clarity would be helpful.

We have revised the abstract methods to read, “Serial CSF samples were obtained from 6 patients, 5 with 2 infections and 1 with 3 infections; the study was limited to those for which CSF samples were available from the end of infection and beginning of reinfection.”

We included the following statement in the Study Subjects section of the Methods as well: “We analyzed serial CSF samples from 6 patients, 5 with 2 infections and 1 with 3 infections; the study was limited to those for which CSF samples were available from the end of infection and beginning of reinfection.”

32. Reviewer #2: 6. For the discussion, it would be helpful to provide more comparative information of the results from the current study with the types of microbes frequently found associated with CSF infections/reinfections, based on the literature. Since there are over 2000 cases every year (assuming in US??), one would think that there would be information regarding the types of microbes associated with these infections. Are there aerobic bacteria often found?

Thank you. We have added a statement: “The most common pathogens recovered from conventional microbiologic cultures include primarily Staphylococcus epidermidis and Staphylococcus aureus, followed by gram-negative organisms.[ref]”

We appreciate the suggestions and look forward to your response.

Tamara Simon

---

## [Decision Letter · Decision Letter 1]

15 Dec 2020

Characterization of cerebrospinal fluid (CSF) microbiota from patients with CSF shunt infection and reinfection using high throughput sequencing of 16S ribosomal RNA genes

PONE-D-19-35359R1

Dear Dr. Simon,

We’re pleased to inform you that your manuscript has been judged scientifically suitable for publication and will be formally accepted for publication once it meets all outstanding technical requirements.

Kind regards,

Brenda A Wilson, Ph.D.

Academic Editor

PLOS ONE

Additional Editor Comments (optional):

Reviewers' comments:

Reviewer's Responses to Questions

**Comments to the Author**

1. If the authors have adequately addressed your comments raised in a previous round of review and you feel that this manuscript is now acceptable for publication, you may indicate that here to bypass the “Comments to the Author” section, enter your conflict of interest statement in the “Confidential to Editor” section, and submit your "Accept" recommendation.

Reviewer #1: All comments have been addressed

Reviewer #2: All comments have been addressed

2. Is the manuscript technically sound, and do the data support the conclusions?

Reviewer #1: Yes

Reviewer #2: Yes

3. Has the statistical analysis been performed appropriately and rigorously? 

Reviewer #1: Yes

Reviewer #2: Yes

4. Have the authors made all data underlying the findings in their manuscript fully available?

Reviewer #1: Yes

Reviewer #2: Yes

5. Is the manuscript presented in an intelligible fashion and written in standard English?

Reviewer #1: Yes

Reviewer #2: Yes

6. Review Comments to the Author

Reviewer #1: Authors have performed additional analyses and addressed most of the critical concerns/comments raised by the reviewer. I have no additional comments.

Reviewer #2: The revised manuscript is much improved after addressing the reviewer comments.

7. PLOS authors have the option to publish the peer review history of their article (what does this mean?). If published, this will include your full peer review and any attached files.

Reviewer #1: No

Reviewer #2: No

---

## [Editor Report · Acceptance letter]

22 Dec 2020

PONE-D-19-35359R1 

Characterization of cerebrospinal fluid (CSF) microbiota from patients with CSF shunt infection and reinfection using high throughput sequencing of 16S ribosomal RNAgenes 

Dear Dr. Simon:

I'm pleased to inform you that your manuscript has been deemed suitable for publication in PLOS ONE. Congratulations! Your manuscript is now with our production department. 

Kind regards, 

on behalf of

Dr. Brenda A Wilson 

Academic Editor

PLOS ONE